# C-phycoerythrin from *Phormidium persicinum* Prevents Acute Kidney Injury by Attenuating Oxidative and Endoplasmic Reticulum Stress

**DOI:** 10.3390/md19110589

**Published:** 2021-10-20

**Authors:** Vanessa Blas-Valdivia, Plácido Rojas-Franco, Jose Ivan Serrano-Contreras, Andrea Augusto Sfriso, Cristian Garcia-Hernandez, Margarita Franco-Colín, Edgar Cano-Europa

**Affiliations:** 1Laboratorio de Neurobiología, Departamento de Fisiología, Escuela Nacional de Ciencias Biológicas, Instituto Politécnico Nacional, Ciudad de México 07738, Mexico; vblasv@ipn.mx (V.B.-V.); cgarciah1702@alumno.ipn.mx (C.G.-H.); 2Laboratorio de Metabolismo I, Departamento de Fisiología, Escuela Nacional de Ciencias Biológicas, Instituto Politécnico Nacional, Ciudad de México 07738, Mexico; projasf@ipn.mx; 3Department of Metabolism, Digestion and Reproduction, Division of Systems Medicine, Section of Biomolecular Medicine, Faculty of Medicine, Imperial College London, South Kensington Campus, London SW7 2AZ, UK; j.serrano-contreras@imperial.ac.uk; 4Department of Chemical and Pharmaceutical Sciences, University of Ferrara, 44121 Ferrara, Italy; sfrndr@unife.it

**Keywords:** C-phycoerythrin, *Phormidium persicinum*, acute kidney injury, mercury, oxidative stress, endoplasmic reticulum stress

## Abstract

C-phycoerythrin (C-PE) is a phycobiliprotein that prevents oxidative stress and cell damage. The aim of this study was to evaluate whether C-PE also counteracts endoplasmic reticulum (ER) stress as a mechanism contributing to its nephroprotective activity. After C-PE was purified from *Phormidium persicinum* by using size exclusion chromatography, it was characterized by spectrometry and fluorometry. A mouse model of HgCl_2_-induced acute kidney injury (AKI) was used to assess the effect of C-PE treatment (at 25, 50, or 100 mg/kg of body weight) on oxidative stress, the redox environment, and renal damage. ER stress was examined with the same model and C-PE treatment at 100 mg/kg. C-PE diminished oxidative stress and cell damage in a dose-dependent manner by impeding the decrease in expression of nephrin and podocin normally caused by mercury intoxication. It reduced ER stress by preventing the activation of the inositol-requiring enzyme-1α (IRE1α) pathway and avoiding caspase-mediated cell death, while leaving the expression of protein kinase RNA-like ER kinase (PERK) and activating transcription factor 6α (ATF6α) pathways unmodified. Hence, C-PE exhibited a nephroprotective effect on HgCl_2_-induced AKI by reducing oxidative stress and ER stress.

## 1. Introduction

Acute kidney injury (AKI), a syndrome engendered by sepsis, cardiorenal syndrome, urinary tract obstruction, and nephrotoxins, is known to increase the level of serum creatinine and/or decrease urine output. It is an important public health issue because of being a serious complication for 10–15% of hospitalized patients and ~50% of those in intensive care [1].

Animal models of AKI are induced by administering a drug or toxicant (e.g., HgCl_2_) [2,3]. Mercury targets the kidney by binding to thiol-containing proteins in the tubular and glomerular nephron portion, disrupting the tubular transport mechanism related to Na^+/^K^+^-ATPase [4]. It also alters the intracellular calcium current and consequently the redox environment. The increase in oxidants is not counteracted by the antioxidant system and therefore leads to oxidative stress [5,6] and endoplasmic reticulum (ER) stress [3].

ER stress disrupts proteostasis in this organelle, causing the accumulation of unfolded and misfolded proteins. To maintain ER function, the unfolded protein response is activated through the protein kinase RNA-like ER kinase (PERK), activating transcription factor 6α (ATF6α), and inositol-requiring enzyme 1α (IRE1α) pathways.

The PERK pathway, crucial in regulating the unfolded protein response, reduces transcription through phosphorylation of the eukaryotic translation initiation factor-2α (eIF2α). If ER stress is controlled, protein folding can resume, and the phosphorylated eIF2α dephosphorylates. In the event that ER stress is sustained, the activating transcription factor 4 (ATF4)/growth arrest and DNA damage-inducible gene 153 (GADD153, also called CHOP) pathway activates the expression of genes that participate in redox homeostasis, autophagy, and/or apoptosis. The particular genes involved depend on the level of ER stress [7,8].

In parallel, the ATF6α pathway diminishes ER stress by regulating genes that encode ER chaperones and enzymes responsible for promoting folding, maturation, secretion, or degradation of misfolded proteins. When ER stress is sustained, the cell activates autophagy and apoptosis by upregulating the generation of reactive oxygen species (ROS) and activating ER membrane-associated caspase 12 through the ATF4/GADD153 pathway [7,8].

Additionally, IRE1α contributes to adaptation or apoptosis under chronic ER stress. The adaptation response of IRE1α is activated by selective cleavage of X-box binding protein 1 (XBP1) mRNA to produce spliced isoforms of XBP1, which enhance the transcription of chaperones, foldases, and components of the ER-associated protein degradation (ERAD) response to restore proteostasis. In case ER stress is still uncontrolled, IRE1α activates c-Jun N-terminal kinases 1 (JNK1) to promote the translocation of B-cell lymphoma 2 (Bcl-2)-associated X protein (Bax) into the mitochondrial membrane, which triggers the release of cytochrome c and the second mitochondrial-activated factor (Smac), leading to the activation of caspases 3 and 9 [9].

Some research groups have been developing eco-friendly therapeutic strategies for AKI from microalgae pigments such as phycobiliproteins, toroidal light-harvesting proteins in cyanobacteria, and the photosynthetic apparatus in algae. The most studied phycobiliprotein with nephroprotective activity is C-phycocyanin (C-PC). It impedes kidney failure by decreasing oxidative stress and ER stress in mice intoxicated with mercury [6,10,11]. Moreover, other phycobiliproteins, including C-phycoerythrin (C-PE), have nutraceutical activity against metabolic and toxic injury that affects certain organs (e.g., the liver) in animal models [12,13].

C-PE, an oligomeric chromoprotein of cyanobacteria, is composed of monomers αβ and prosthetic covalently linked open-chain tetrapyrrole moieties denominated C-phycoerythrobilin. In *Phormidium* sp., the monomer units oligomerize to form trimers (αβ)_3_ and then stack as hexamers [(αβ)_3_]_2_ [14]. C-PE is widely used in the food and cosmetic industries, as well as in diagnosis and research. There are reports on its nutraceutical properties, which stem from scavenging and antioxidant activity [15]. Our research group demonstrated that a C-PE-rich protein extract from *Pseudanabaena tenuis* has nephroprotective activity [16], although its mechanism is still not completely understood. The aim of the current contribution was to determine whether the nephroprotective activity of C-PE (purified from *Phormidium persicinum*) is related to a reduction in oxidative stress and ER stress, and consequently an attenuation of the alterations in the levels of nephrin and podocin normally caused by HgCl_2_-induced AKI.

## 2. Results

### 2.1. Characterization of C-PE from Phormidium persicinum

The absorbance spectra from various steps of purification (Figure 1) show an absorbance peak at 562 nm. The A_562_/A_280_ ratio increased with each purification step, thus being the greatest (4.35) for the final product of purified C-PE.

The images of native- and SDS-PAGE at each step of the purification process show that the α and β C-PE subunits correspond to ~19 and ~21 KDa, respectively (Figure 2).

The excitation-emission matrix (EEM) spectrum corresponding to the 3D fluorescence fingerprint of purified C-PE is shown in Figure 3 (panel A). The expansion of the same EEM displays the emission and excitation regions in the range of 555–595 and 510–570 nm, respectively (panel B). The fingerprint of C-PE exhibits a sharp fluorescence peak at E_ex_/E_em_ 563/574 nm (corresponding to fluorochrome) next to Rayleigh-Tyndall’s scattered light lines. The 3D spectrum of EEM features three principal Ex/Em peaks at 563/574, 545/574, and 530/574, and a small Ex/Em peak at 385/575. Two shoulders are present on the lower part of the main peak, the first at E_ex_/E_em_ 545/574 nm and the second at E_ex_/E_em_ 530/574 nm. Another weak peak can be observed at E_ex_/E_em_ 385/575 nm.

### 2.2. Evaluation of Oxidative Stress, the Redox Environment, the Activity of Effector Caspases 3 and 9, the Expression of Nephrin and Podocin, and Renal Damage

The effect of C-PE on HgCl_2_-induced oxidative stress and alterations in the redox environment is illustrated in Figure 4 (panels A–C and D–E, respectively). Animals intoxicated with HgCl_2_ showed higher renal oxidative stress, indicated by the corresponding increase in lipid peroxidation (panel A, ~374%), ROS (panel B, ~211%), and nitrites (panel C, ~171%). Mercury intoxication also caused a lower GSH^2^/GSSG ratio (panel F, ~66%) and greater GSSG content (panel E, ~269%). On the other hand, all doses of C-PE treatment prevented the HgCl_2_-induced increase in lipid peroxidation, ROS, and GSSG, and the alteration in the GSH^2^/GSSG ratio, while ameliorating the elevated level of nitrites (from 171% to 139%).

Regarding the proteins associated with glomerular damage (Figure 5), mercury decreased the expression of nephrin (A) and podocin (B) by ~65% and ~71%, respectively. Treatment with C-PE partially reduced, by ~36% and ~48%, the downregulation of nephrin and podocin, respectively. These changes can be appreciated by the corresponding Western blots (Figure 5C).

According to typical photomicrographs of the renal cortex stained with hematoxylin-eosin (H&E) (Figure 6), the control (vehicle only) and C-PE only groups had normal cytoarchitecture, which is characterized by glomeruli and the surrounding tubules with cuboidal epithelium. The photomicrographs of the group treated with mercury only display edema, cellular atrophy of distal and proximal tubules, distortion of cellular continuity, loss of the cell nucleus, hyperchromatic nuclei, and glomerulosclerosis. The AKI mice treated with C-PE exhibited a dose-dependent nutraceutical effect capable of preventing cellular damage.

The effect of C-PE on the activity of caspases 3 and 9 is shown in Figure 7 (panels A and B, respectively). HgCl_2_ generated an increase of ~511% and ~347% in the level of caspases 3 and 9, respectively. These results indicate grade 4 histological damage (panel C), affecting over 75% of the tubules and glomerulus. C-PE diminished damage in a dose-dependent manner (panel C). The highest C-PE dose (100 mg/kg/day) led to grade 1–2 kidney damage, affecting 25–50% of the tubules and glomerulus.

### 2.3. Evaluation of ER Stress

The effects of C-PE on the PERK/p-eIF2α (Ser52)/ATF4 and PERK/p-eIF2α (Ser52)/ATF6α signaling pathways is portrayed in Figure 8. HgCl_2_-induced AKI was manifested as an overexpression of PERK (A), p-eIF2α (Ser 52) (B), ATF4 (C), GADD153 (D), GADD34 (E), and ATF6α (F). The C-PE treatment did not prevent the alteration in the expression of these proteins in both pathways. A representative Western blot of the marker for the PERK/eIF2α/ATF4 and PERK/eIF2α/ATF6α signaling pathways is shown in Figure 9.

Figure 10 shows the effect of C-PE on the IRE1α pathway and the proteins associated with cellular damage. HgCl_2_ exposure generated an overexpression of IRE1α (panel A), XBP1 (panel B), caspase 12 (panel C), Bax (panel D), p-p53 (Thr 155) (panel G), and p53 (panel H). It also increased the Bax/Bcl2 and p-p53 (Thr 155)/p53 ratios (panels F and I, respectively) and reduced the expression of Bcl2 (panel E). With C-PE treatment, there was no alteration in the level of any of the proteins evaluated, which is observed in the corresponding Western blot depicted in Figure 11.

## 3. Discussion

C-PE is reported to have nutraceutical activity against the damage resulting from cell insult [12,13]. Our group has demonstrated that treatment with a protein extract rich in C-PE prevented oxidative stress and cellular damage in an animal model of HgCl_2_-induced AKI [16]. This model was chosen because mercury produces ER stress, which leads to renal damage. However, the aforementioned study only associated the nutraceutical properties of C-PE with scavenging and antioxidant activity. Thus, the aim of the current contribution was to explore the molecular mechanism of action of C-PE (purified from *P. persicinum*) by examining its nephroprotective activity against HgCl_2_-induced ER stress, oxidative stress, and alterations in the redox environment in the same animal model.

HgCl_2_ produces oxidative stress and alterations in the redox environment by three mechanisms: Fenton and Haber-Weiss reactions that generate free radicals and ROS [17], the activation of ER stress [3], and the binding of Hg^2+^ with intracellular sulfhydryl-containing proteins and low-molecular-weight compounds (e.g., GSH) capable of affecting the redox environment and protein function [18]. As a consequence of these reactions, nephrin and podocin are downregulated, and the slit diaphragm is injured, which is observed as HgCl_2_-induced AKI. The resulting inflammatory process participates in the progression of AKI [19].

In recent years, the use of nutraceuticals from cyanobacteria and their metabolites has proven effective against renal damage (e.g., AKI) stemming from toxicants or chronic kidney disease [16,20,21,22]. Purified C-PE presently demonstrated nephroprotective activity when tested against HgCl_2_-induced AKI, as evidenced by the reduction found in oxidative stress and ER stress.

C-PE, a protein with a molecular weight of ~240 KDa, has nutraceutical properties in vitro as an ROS scavenger [23]. Moreover, it prevents oxidative stress and cellular damage in vivo [12,13]. All reports on C-PE suggest that it is a potent antioxidant. By scavenging ROS, it avoids alterations in the redox environment and therefore impedes cellular damage [12,13,24]. However, animal studies have not yet completely defined the nutraceutical protection mechanism.

C-PE may act as a prodrug that leads to the release of the phycoerythrobilin moiety into the gastrointestinal tract, as previously demonstrated by our group for C-PC and phycocyanobilin [22]. C-PC is known to break down into chromo-peptides that contain phycocyanobilin, followed by the apparent absorption of linear tetrapyrrole compounds facilitated by the action of intestinal peptidases [24,25]. Once in serum, phycoerythrobilin could bind to albumin due to its low water solubility, which would extend its therapeutic activity into the entire organism [26].

The protective effect of C-PE against HgCl_2_-induced AKI is associated with antioxidant, anti-inflammatory, and chelation mechanisms. C-PE acts as an antioxidant because it contains PEB. In addition, the chemical structure of phycoerythrobilin acts as a nucleophilic compound, neutralizing free radicals and ROS [24]. According to an in vitro model, the chelation of Hg^2+^ by PEB suppresses the degranulation of RBL-2H3 mast cells and decreases the intracellular concentration of Ca^2+^ [27], giving rise to anti-inflammatory and nephroprotective effects. Hg^2+^ binds to PEB thioether bridges in C-PE, which assume a cyclic helical form capable of chelation [28]. The antioxidant and chelating activity of C-PE can avoid Fenton and Haber-Weiss reactions and consequently ameliorate the production of free radicals, the generation of oxidative stress, and the alteration of the redox environment in kidney cells. All the aforementioned mechanisms of C-PE are related to the maintenance of the redox environment and therefore prevent the dysfunction of organelles such as the ER.

In the current evaluation of proteostasis, HgCl_2_-induced ER stress was found to activate the IRE1α pathway and promote cell death. At the same time, mercury activated the PERK pathway, which restored proteostasis through PERK/eIF2α/ATF-4/GADD153. When the cell was incapable of compensating for imbalances in proteostasis, the activation of ATF4 and GADD153 in the same pathway led to the expression of proapoptotic proteins and the triggering of cell death. As can be appreciated, PERK and IRE1α have a synergic effect in prompting kidney cell death by increasing the Bax/Bcl-2 ratio and the level of caspases 3, 8, 9, and 12 [3,10]. Hence, HgCl_2_ was capable of generating AKI in the present study by fomenting oxidative stress, an alteration in the redox environment, and ER stress. The resulting histological damage was considerable (grade 4), affecting over 75% of tubular and glomerular cells.

C-PE treatment enhanced the canonical ER response through the PERK/p-eIF2α (ser 52)/ATF-4/GADD153 pathway, involving ER-associated degradation (ERAD), known to process misfolded and unfolded proteins. The phosphorylation of eIF2α (ser 52) is able to suppress the overall translation of mRNA, thus reducing protein stress in the ER. Furthermore, the moderate increment in ATF6α upregulates several genes that participate in the adaptative phase of the unfolded protein response [29]. C-PE treatment is herein proposed to have activated the PERK and ATF6 signaling pathways, maintaining proteostasis by avoiding oxidative stress and alterations in the redox environment and by activating the unfolded protein response [30,31].

The response elicited by C-PE is distinct from that of other phycobiliproteins. For instance, C-PC averts the overexpression of GADD34 by activating GADD153, which is related to the inhibition of apoptosis [11,32]. On the other hand, both C-PC and C-PE maintain proteostasis. The differences between these two responses should be explored in depth in future research.

C-PE and C-PC have a similar effect on the IREα pathway, decreasing cell death mediated by caspases 3, 9, and 12 as well as reducing the disruption in p53 activation and the alteration of the Bax/Bcl2 ratio [10,11]. This idea is supported by neurotoxicological models, where C-PE prevents ER stress linked to calcium deregulation and mitochondrial dysfunction [33].

In the control group, interestingly, C-PE per se increased the phosphorylation of p53 (Thr 155), which is a genome gatekeeper because it is a master transcriptional factor that induces cellular senescence and suppresses cell growth and tumor formation. Exposure to various cellular stressors, however, causes p53 to be overexpressed and phosphorylated in several regions, leading to cell cycle arrest or apoptosis. Accordingly, p53 is phosphorylated by the C-Jun activation domain-binding protein-1 (Jab1) in Thr 155, promoting its translocation into the cytoplasm to favor interaction with the COP9 signalosome complex. These nuclear export mechanisms of p53 provide a practical future approach to a possible C-PE-induced activation of anti-cancer therapy by p53 [34], as evidenced by the lack of histological irregularities in the C-PE control group as well as the capacity of C-PE treatment of AKI mice to prevent oxidative stress, ER stress, and alterations in the redox environment and cell death markers.

## 4. Materials and Methods

### 4.1. Animals

Forty-eight male albino NIH Swiss mice (25–30 g) were kept in a cool room (21 ± 2 °C) with 40–60% relative humidity under a 12/12 h light/dark cycle (lights on at 8 AM). Food and water were provided ad libitum. The experimental procedures were in accordance with the Official Mexican Norm (NOM-062-ZOO-1999, technical specifications for the production, care, and use of laboratory animals) [35]. The protocol was approved by the institutional Internal Bioethics Committee (ZOO-013-2021).

The animals were divided into two lots to carry out distinct protocols, one to assess oxidative stress and kidney damage and another to analyze ER stress. For the evaluation of oxidative stress and kidney damage, 36 mice were randomly allocated to 6 groups (*n* = 6). Three were control groups: (1) the vehicle (negative control), with 100 mM of phosphate buffer (PB, at pH 7.4) administered by oral gavage (og) + 0.9% of saline solution (SS) applied intraperitoneally (ip), (2) AKI induced by a single application of 5 mg/kg HgCl_2_ ip + the vehicle (PB) og, and (3) C-PE treatment, consisting of 100 mg/kg/day C-PE og + 0.9% SS ip. The other three groups received a single application of HgCl_2_ ip as well as 25, 50, or 100 mg/kg/day C-PE og. For the analysis of ER stress, twelve mice were randomly allocated to four groups with the following treatments (*n* = 3): (1) the control (vehicle), (2) mercury-induced AKI, (3) the C-PE treatment, and (4) the AKI + C-PE treatment (a single application of HgCl_2_ ip and 100 mg/kg/day C-PE og).

C-PE or the vehicle was administered 30 min before the injection of HgCl_2_ or 0.9% of SS. C-PE was administered once daily for five days (the first protocol) or for three days (the second protocol) at the same time (12:00 AM) each day. Whereas the mice assigned to the evaluation of oxidative stress and renal damage were euthanized 5 days after mercury intoxication, those employed for assessing ER stress were euthanized 3 days after the same event. The right kidneys were frozen at −70 °C to await examination of the markers of oxidative stress and the redox environment by Western blot, while the left kidneys were put into paraformaldehyde in PBS (4% *v/v*) to appraise cell damage.

### 4.2. Cultivation, Purification, and Characterization of C-PE from Phormidium persicinum

*P. persicinum* was obtained from the culture collection of the Centro de Investigaciones Biológicas del Noroeste, S. C. (CIB 84). It was grown in a synthetic medium (denominated NM), created and optimized by our group (composition: 29 g/L of commercial sea salt, 0.8 g/L NaHCO_3_, 0.05 g/L K_2_HPO_4_, 2.16 g/L NaNO_3_, 5 mg/L MgSO_4_, 1 mg/L FeSO_4_, and 1 mL of a micronutrient solution containing 0.2 mM EDTA, 46.2 mM H_3_BO_3_, 9.3 mM MnCl_2_, 0.95 mM ZnSO_4_, 2.03 mM Na_2_MoO_4_, 0.49 mM Ca(NO_3_)_2_, and 0.77 mM CuSO_4_). Incubation was carried out at 21 ± 2 °C with constant aeration provided by an air pump, under green LED illumination (24 W, 3000 Lx) and a 12/12 h light/dark cycle (lights on at 8:00 AM).

Regarding the purification of C-PE, the cyanobacterial biomass was centrifuged at 10,000× *g* for 1 min and 5–10 g of the resulting cell pellet was re-suspended in 20 mL of distilled water. Subsequently, three freeze–thaw cycles were performed, freezing at −20 °C and thawing at 4 °C during 24 h. The resulting slurry was centrifuged in 4 cycles at 21,400× *g* for 10 min at 4 °C to remove the cell debris. An aliquot of 20 mL of the phycobiliprotein-rich extract was injected into a column (33 cm long × 4.7 cm in diameter) containing Sephadex G-250 gel previously equilibrated with 10 mM of PB (pH 7.4). The pink fractions were obtained and precipitated with a saturated solution of (NH_4_)_2_SO_4_ at 4 °C for 24 h in the dark. This mixture was centrifuged at 21,400× *g* for 2 min at 4 °C, and the resulting pellet was resuspended in 100 mM of PB at pH 7.4. The membrane was then dialyzed with PB for 24 h, after which time an aliquot of C-PE was immediately lyophilized to construct a calibration curve, obtain an absorption spectrum, and characterize the extract fluorometrically with an EEM. The C-PE extract was solubilized in PB and 5 mM of sucrose and frozen at −20 °C to await administration to the animals [36].

The EEM was recorded by scanning excitation and emission simultaneously in a Luminescent Spectrometer (Perkin Elmer LS 55) equipped with a Xenon discharge lamp and an excitation/emission slit 5/5. The scans were processed by 3D View Perkin Elmer software to produce 3D fingerprint contour maps by using fluorescence lines (with emission plotted on the *X*-axis and excitation on the *Y*-axis), as previously reported [37].

The calibration curve of 0.6–6 mg/mL of C-PE solubilized in PB was calculated as follows:$$CPE\;\left(\frac{mg}{dL}\right)=\frac{\left[Absorbance_{562\;nm}-0.1374\right]}{0.3540};r^{2}=0.9899;r=0.9949;\;\in Absorbance_{562\;nm\;}\left[0.330-2.245\right]$$

The purity index was calculated as the ratio of the maximum absorbance peak to the absorbance peak of the proteins (A_562_/A_280_) [38].

### 4.3. Evaluation of Oxidative Stress, the Redox Environment, and the Activity of Effector Caspases 3 and 9

Kidneys were homogenized in 3.5 mL of 10 mM PB for all assays. The quantification of oxidative stress, the redox environment, and the activity of effector caspases 3 and 9 was performed with a previously described method [3,22].

The lipid peroxidation technique employed an aliquot of 500 µL of homogenate, which was added to 4 mL of chloroform-methanol (2:1, *v/v*). The mixture was agitated and kept at 4 °C for 30 min (protected from light) to allow for the separation of the polar and nonpolar phases. Afterwards, the aqueous phase was aspirated and discarded. With an aliquot of 2 mL of the organic phase (chloroform), fluorescence was determined at 370 nm (excitation) and 430 nm (emission). The results were expressed as relative fluorescence units (RFU) per mg of protein.

The level of ROS was quantified by the formation of 2,7-dichlorofluorescein (DCF), and 10 μL of the homogenate was added to 1945 µL of TRIS-HEPES (18:1 *v/v*) and incubated in the presence of 50 µL of 2,7-dichlorofluorescin diacetate (DCFH-DA) at 37 °C for 1 h. The reaction was stopped by freezing, and the fluorescence was measured at 488 nm (excitation) and 525 nm (emission).

Nitrites were assessed as indirect markers of nitrergic stress. An aliquot of 500 µL of homogenate was added to 500 µL of concentrated chlorohydric acid and 500 µL of 20% zinc suspension. The mixture was stirred and incubated at 37 °C for 1 h, followed by centrifugation at 4000× *g* for 2 min. The supernatant (50 µL) was added to a 96-well polystyrene plate containing 50 µL of 0.6% sulfanilamide and 0.12% *N*-(naftyl)-ethylenediamine, and then incubated for 15 min at room temperature. The absorbance was measured at 530 nm in a Multiscan Go^®^ plate spectrophotometer.

A determination was made of two redox environment markers, GSH and GSSG, in a sample of 300 µL, treated with 500 µL of 30% phosphoric acid and centrifuged at 10,000× *g* for 30 min at 4 °C. To analyze GSH, an aliquot of 30 µL of the supernatant was diluted in 1.9 mL of FEDTA (1:10, 100 mM phosphate and 5 mM EDTA), and the mixture was reacted with 100 µL of o-phthaldialdehyde. To assess GSSG, 130 µL of the supernatant was added to 60 µL of N-ethylmaleimide and left for 30 min. Subsequently, an aliquot of 60 µL of the mixture was combined with 1.84 mL of FEDTA and 100 µL of o-phthaldialdehyde. The two chemical species were measured at 350 nm (excitation) and 420 nm (emission).

The activity of caspases 3 and 9 was evaluated using a commercial colorimetric assay kit as specified in the manufacturer’s instructions (Millipore, APT165 and APT173, respectively). Accordingly, *p*-nitroaniline (*p*NA) was cleaved from the substrate *N*-Acetyl-Asp-Glu-Val-Asp *p*-nitroaniline (DEVD-*p*NA, caspase 3) or *N*-Acetyl-Leu-Glu-His-Asp *p*-nitroaniline (LEHD-*p*NA, caspase 9), and then the samples were measured spectrophotometrically at 405 nm.

### 4.4. Examination of Kidney Damage

Histopathological analysis was performed to appraise kidney damage. The kidneys were fixed for 48 h in 4% paraformaldehyde in PBS. Afterwards, they were embedded in paraffin to obtain 5 µm slices on a standard microtome. Each section was stained with H&E, dehydrated, and mounted in resin. The presence or absence of kidney cell damage was evaluated with a histopathological graded scale [6,20,22] as follows: 0, undamaged (indistinguishable from the controls); 1, minimal (affecting ≤25% of the tubules and glomerulus); 2, mild (affecting >25% and ≤50% of the tubules and glomerulus); 3, moderate (affecting >50% and ≤75% of the tubules and glomerulus); 4, severe (affecting >75% of the tubules and glomerulus).

### 4.5. Western Blot Analysis for Nephrin, Podocin, and ER Stress Markers

The expression of proteins was determined with Western blot assays. Briefly, the samples were prepared with 100 µL of the homogenate mixed with 50 μL of a complete protease inhibitor cocktail^®^ (MilliporeSigma, Burlington, MA, USA) in lysis buffer, and then 150 μL of the 2× Laemmli sample buffer (Biorad, Hercules, CA, USA, 161-0737) was added. The samples were homogenized by vortex, placed in a boiling water bath for 3 min, and then kept at −20 °C to await processing. Fifty μg of protein samples were loaded in 15% polyacrylamide gel with sodium dodecyl sulfate (SDS-PAGE) and separated by electrophoresis (90 V for 60 min). Subsequently, the proteins were electrotransferred from the gels to PVDF membranes in a Trans-Blot Turbo system (Biorad) at 25 V and 2.05 A for 7 min. Upon completing this time, the membranes were blocked for 1 h under constant stirring in PBST (PBS with 0.05% Tween 20 and 5% low-fat milkSvelty^®^), followed by incubation overnight at 4 °C in a blocking buffer containing the primary antibodies. The primary antibodies (Santa Cruz Biotechnology, Dallas, TX, USA), diluted 1:1000, were PERK (sc-377400), p-eIF2α (Ser 52, sc-12412), ATF-4 (sc-200), GADD153 (sc-56107), ATF-6α (sc-166659), IRE-1α (sc-390960), p-p53 (Thr 155, sc-377567), XBP1 (sc-7160), Bax (sc-20067), and Bcl2 (sc-7382). Podocin (orb337389), nephrin (orb11107), GADD34 (orb13417), and p53 (orb14498) were acquired from Byorbit (Cambridgeshire, Cambridge, UK) and diluted 1:500. After incubation, membranes were washed three times with fresh PBST (20 min/wash) and then incubated in a secondary antibody diluted 1:1500 (HPR-conjugated goat anti-rabbit; Life Technologies, Rockford, IL, USA, 65-6120) at room temperature for 1 h under constant stirring. Membranes were washed three times with fresh PBST. Finally, the protein bands were revealed on photographic plates (JUAMA, Mexico City, Mexico) by chemiluminescence, using Luminata TM Forte^®^ (MilliporeSigma, Burlington, MA, USA). β-Actin protein expression served as the loading control and constitutive protein (Santa Cruz Biotechnology; sc-1615, dilution 1:4000). The optical density (OD) of all bands was quantified by the Image J program (NIH, Bethesda, MD, USA) and described as the protein/β-actin ratio.

### 4.6. Statistical Analysis

All data are expressed as the mean ± standard error, except for the kidney damage score. The latter is described as the median ± interquartile spaces, with the values analyzed with the Kruskal–Wallis method. The variables in the first protocol (to evaluate oxidative stress and the redox environment) were examined by one-way analysis of variance (ANOVA). Two-way ANOVA was utilized to assess ER stress, considering the treatment and absence/presence of AKI as factors. ANOVA was followed by the Student-Newman-Keuls post hoc test. Statistical significance was considered at *p* < 0.05.

## 5. Conclusions

The nutraceutical effect of C-PE on HgCl_2_-induced AKI stems from its antioxidant activity, which reduces the level of oxidative stress markers and maintains the redox environment. Additionally, C-PE modulates intracellular signaling pathways involved in proteostasis, avoiding the disruption of podocytes and damage to glomerular and tubular cells. Hence, the nephroprotective activity of C-PE is related to the prevention of oxidative stress and ER stress in the kidney of animals intoxicated with mercury. The nutraceutical effect may also be related to anti-inflammatory activity, possibly triggering autophagy as a survival pathway linked to the unfolded protein response. This mechanism is worthy of greater attention in future research.

## Figures and Tables

**Figure 1 marinedrugs-19-00589-f001:**
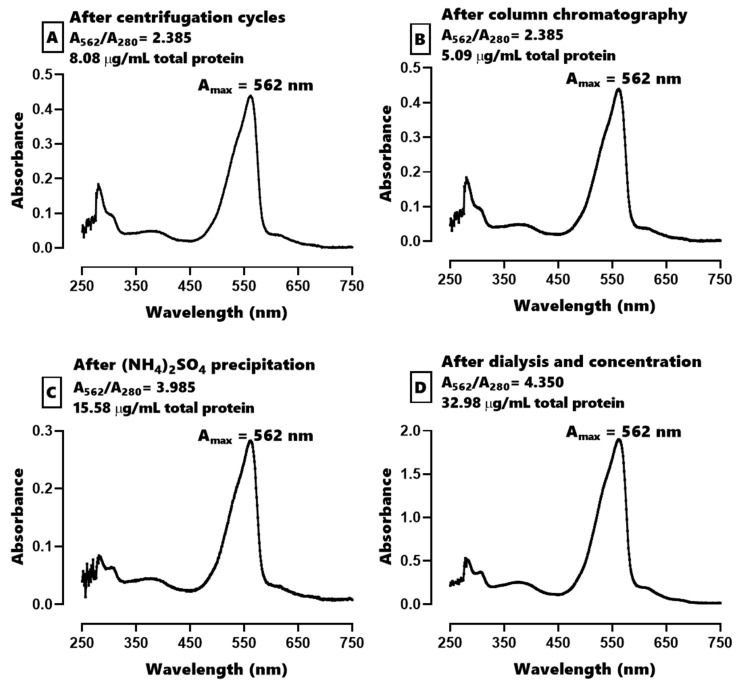
The absorbance spectra for the process of purification of C-phycoerythrin (C-PE) from *Phormidium persicinum* taken after the following events: the centrifugation cycles (**A**), Sephadex column chromatography (**B**), (NH_4_)_2_SO_4_ precipitation (**C**), and dialysis and concentration (**D**).

**Figure 2 marinedrugs-19-00589-f002:**
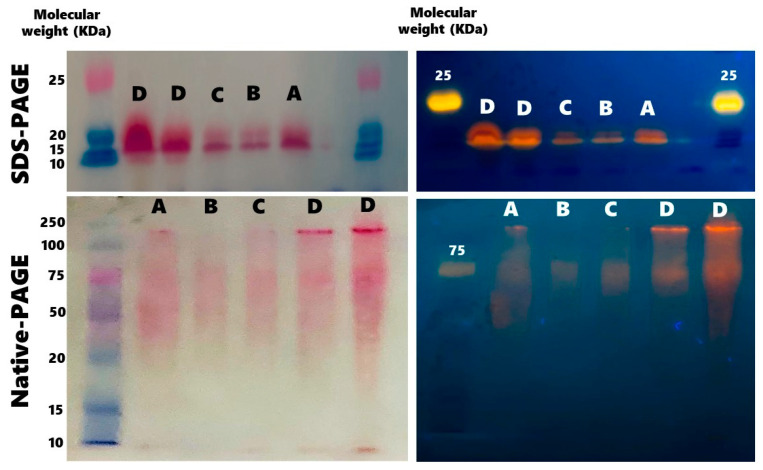
Representative native- and sodium dodecyl sulfate (SDS)-polyacrylamide gels (PAGE) during the process of purification of C-phycoerythrin (C-PE) from *Phormidium persicinum*, taken after the following events: the centrifugation cycles (A), Sephadex column chromatography (B), (NH_4_)_2_SO_4_ precipitation (C), and dialysis and concentration (D).

**Figure 3 marinedrugs-19-00589-f003:**
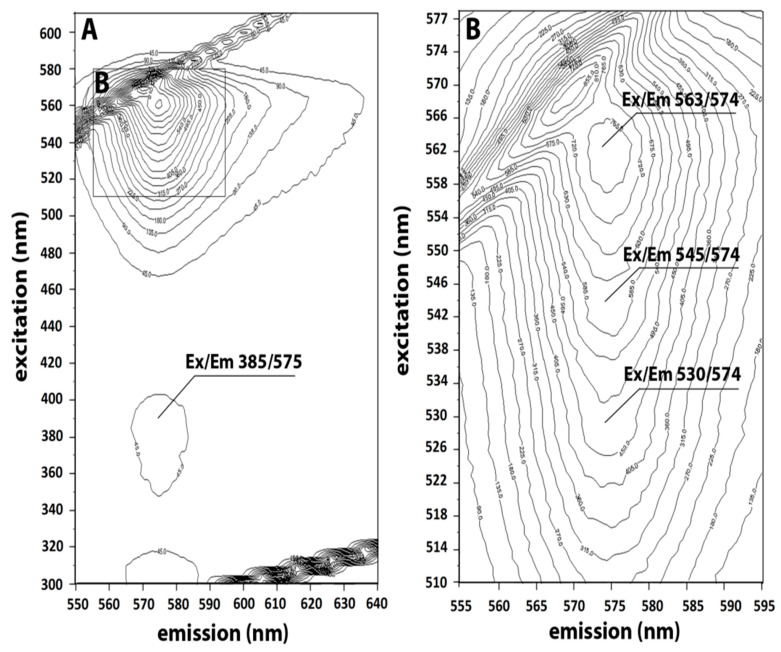
3D spectrum of the excitation-emission matrix (EEM) of C-phycoerythrin (C-PE), with the emission and excitation regions in the range of 550–640 and 300–600 nm, respectively (**A**). Expansion of the EEM for emission (555–595 nm) and excitation (510–577 nm) (**B**).

**Figure 4 marinedrugs-19-00589-f004:**
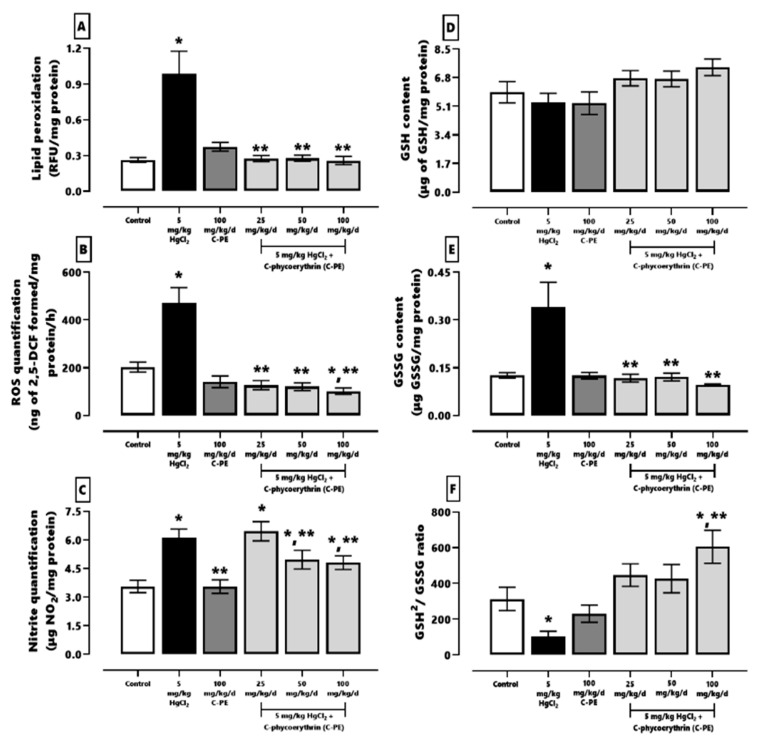
Effect of C-phycoerythrin (C-PE) on HgCl_2_-induced oxidative stress and alterations in the redox environment of the kidney. Oxidative stress markers (**A**–**C**). Redox environment markers (**D**–**F**). Data are expressed as the mean ± SEM (*n* = 6 mice/group). One-way ANOVA and the Student-Newman-Keuls post hoc test. RFU, relative fluorescence units. * *p* < 0.05 vs. the control group. ** *p* < 0.05 vs. the HgCl_2_ group.

**Figure 5 marinedrugs-19-00589-f005:**
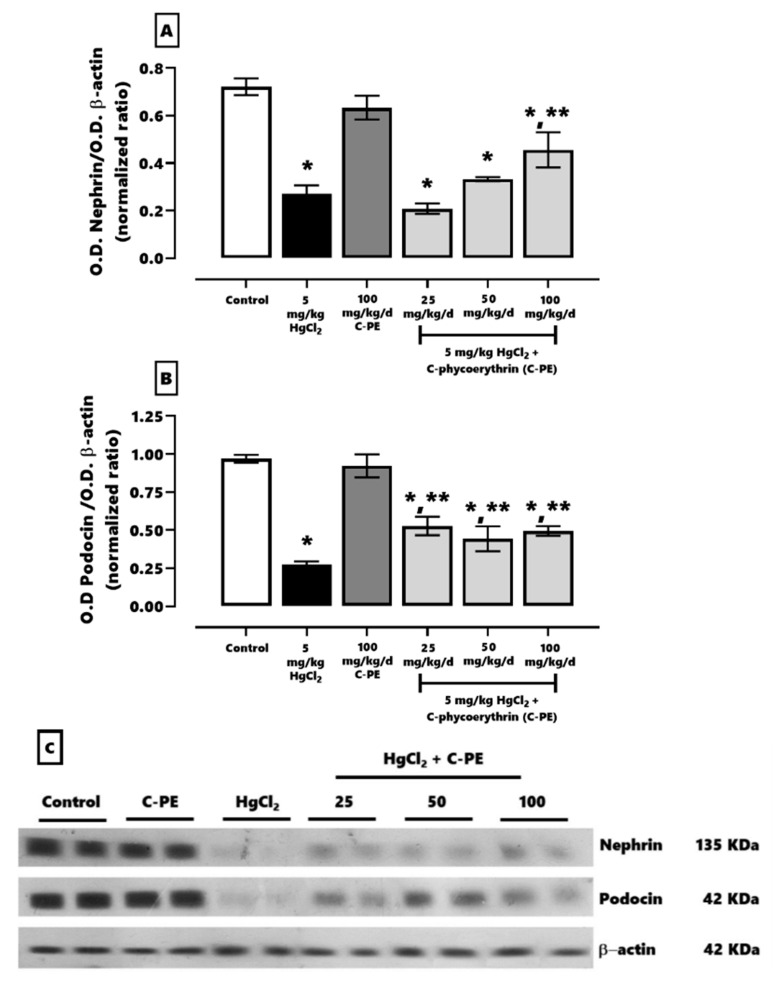
Effect of C-phycoerythrin (C-PE) on the decreased expression of nephrin (**A**) and podocin (**B**) in the kidneys that results from the exposure of mice to HgCl_2_. (**C**) Representative Western blots of each experimental group. Data are expressed as the mean ± SEM (*n* = 6 mice/group). OD, optical density. One-way ANOVA and the Student-Newman-Keuls post hoc test. * *p* < 0.05 vs. the control group. ** *p* < 0.05 vs. the HgCl_2_ group.

**Figure 6 marinedrugs-19-00589-f006:**
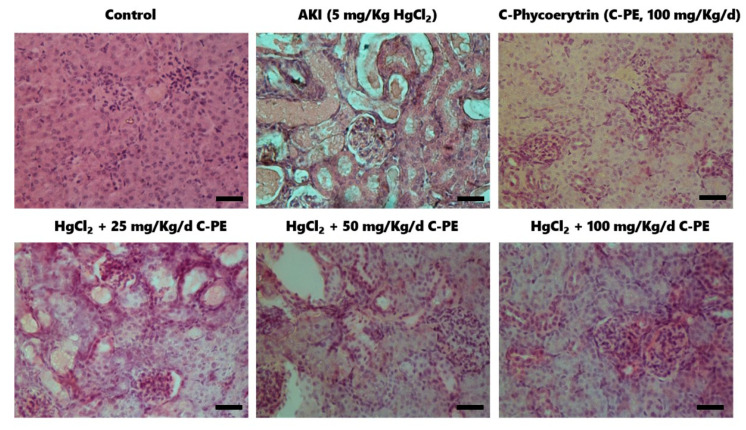
Representative photomicrographs of the renal cortex of animals intoxicated with HgCl_2_ and treated with C-phycoerythrin (C-PE). HgCl_2_ causes cell atrophy, hyperchromatic nuclei, and edema. Histological alterations were ameliorated in groups treated with C-PE. The tissue was stained with hematoxylin-eosin. The lower right bar represents 250 µm.

**Figure 7 marinedrugs-19-00589-f007:**
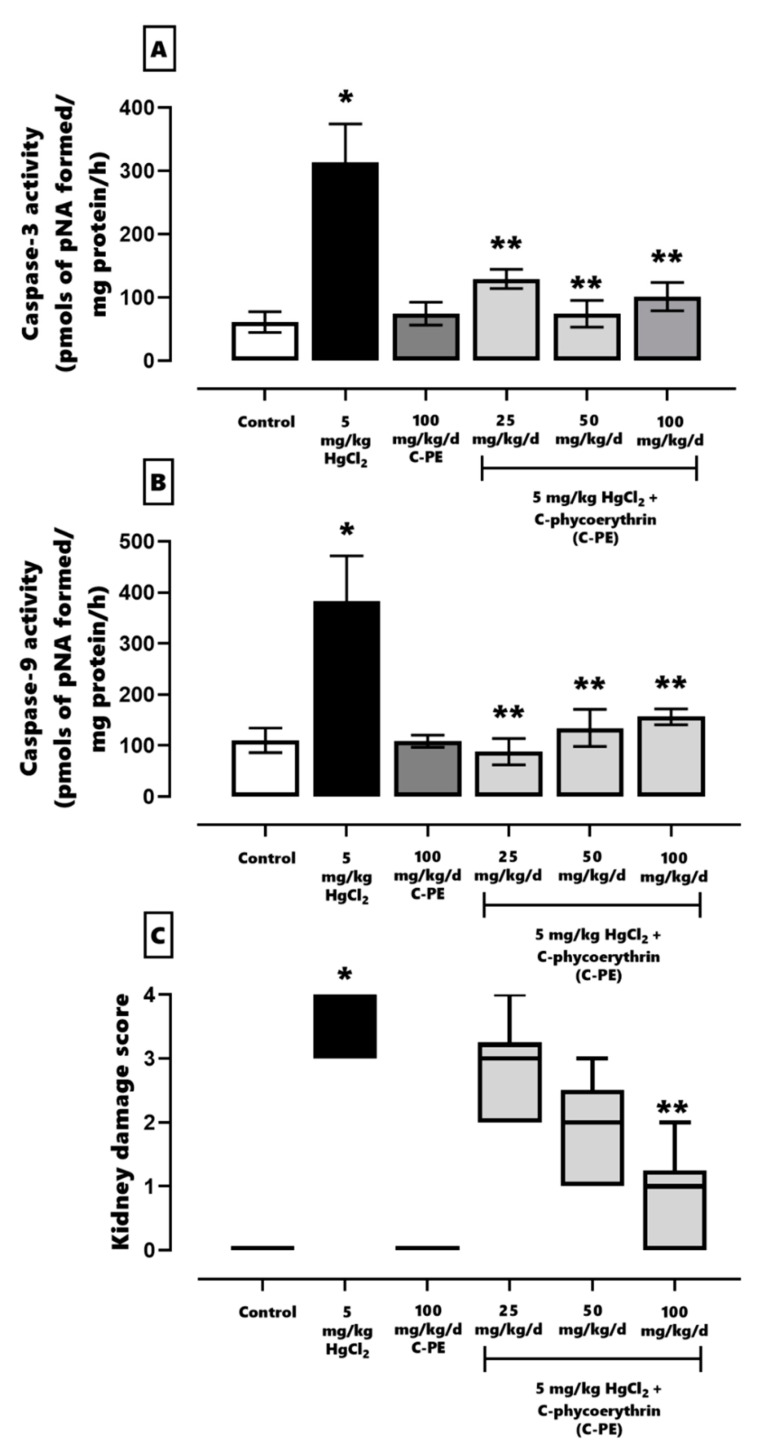
The effect of C-phycoerythrin (C-PE) on the activity of caspases 3 (**A**) and 9 (**B**) and the kidney damage score (**C**) in mice with HgCl_2_-induced AKI. In (**A**) and (**B**), data are expressed as the mean ± SEM (*n* = 6 mice/group). Data were analyzed with one-way ANOVA and the Student-Newman-Keuls post hoc test. * *p* < 0.05 vs. the control group. ** *p* < 0.05 vs. the HgCl_2_ group. In (**C**), each box represents the median ± interquartile space. Data were examined with the Kruskal–Wallis test and Dunn post hoc test. * *p* < 0.05 vs. the control group. ** *p* < 0.05 vs. the HgCl_2_ group.

**Figure 8 marinedrugs-19-00589-f008:**
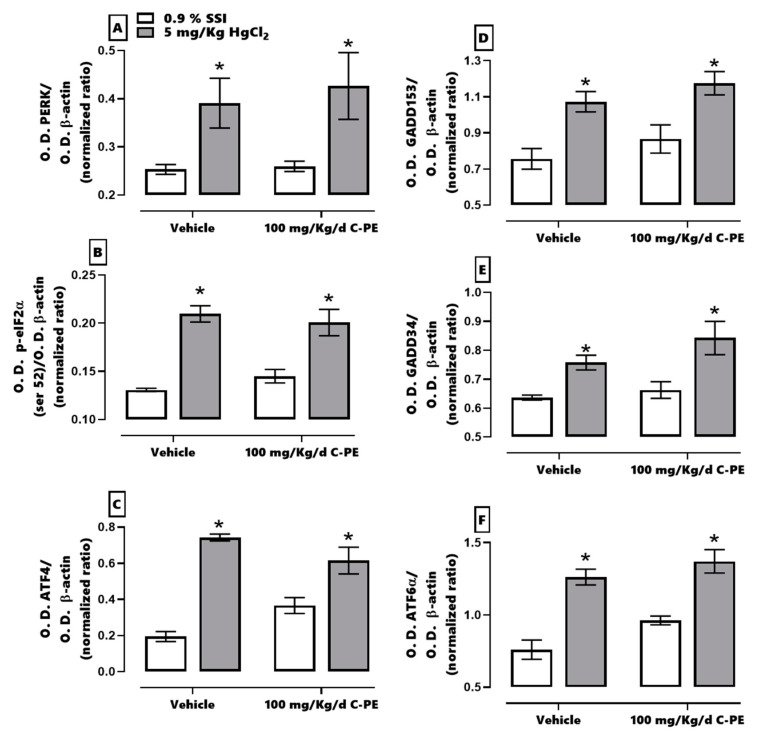
Effect of C-phycoerythrin (C-PE) on HgCl_2_-induced endoplasmic reticulum stress through the PERK/p-eIF2α (Ser 52)/ATF4-GADD153 and PERK/p-eIF2α (Ser 52)/ATF6α/GADD153 pathways in the kidney. An evaluation was made of the expression of PERK (**A**), p-eIF2α (Ser 52) (**B**), ATF4 (**C**), GADD153 (**D**), GADD34 (**E**), and ATF6α (**F**). Data are expressed as the mean ± SEM (*n* = 3 mice/group). OD, optical density. * *p* < 0.05 vs. the control group.

**Figure 9 marinedrugs-19-00589-f009:**
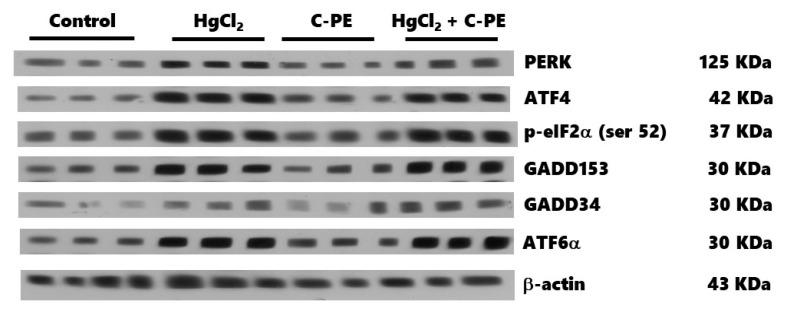
Representative Western blot of the effect of C-phycoerythrin (C-PE) on HgCl_2_-induced endoplasmic reticulum stress through the PERK/p-eIF2α (Ser 52)/ATF4/GADD153 and PERK/p-eIF2α (Ser 52)/ATF6α/GADD153 pathways.

**Figure 10 marinedrugs-19-00589-f010:**
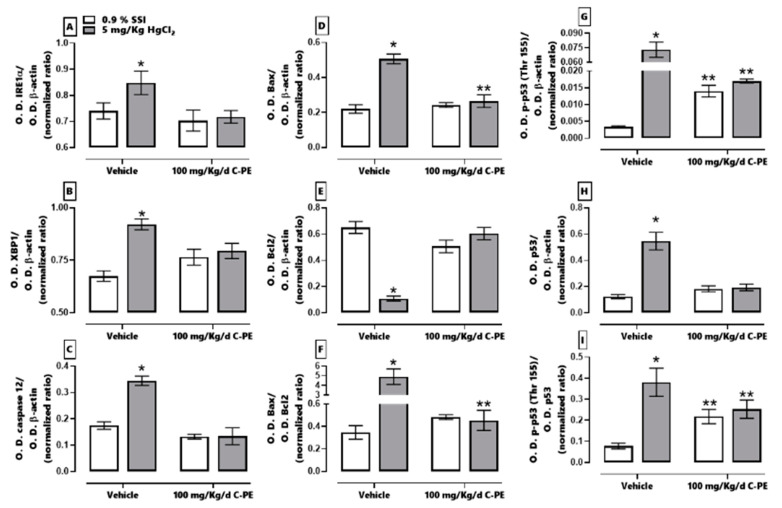
Effect of C-phycoerythrin (C-PE) on HgCl_2_-induced endoplasmic reticulum stress and cell death. Protein expression was evaluated for IRE1α (**A**), XBP1 (**B**), caspase 12 (**C**), Bax (**D**), Bcl2 (**E**), the Bax/Bcl2 ratio (**F**), p53 (**G**), p-p53 (Thr 155) (**H**), and the p53/p-p53 (Thr 155) ratio (**I**). Data are expressed as the mean ± SEM (*n* = 3 mice/group). OD, optical density. * *p* < 0.05 vs. the control group. ** *p* < 0.05 vs. the HgCl_2_ group.

**Figure 11 marinedrugs-19-00589-f011:**
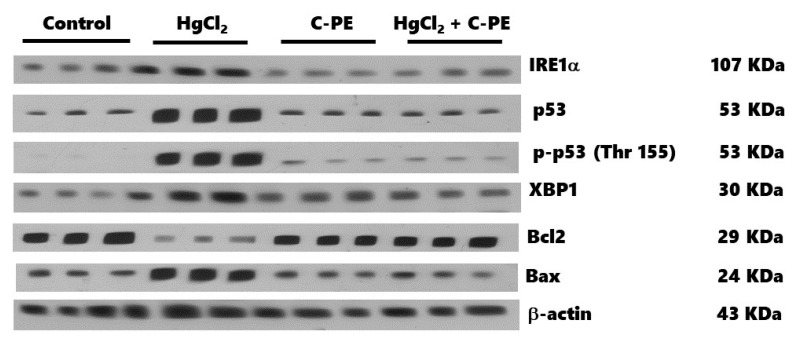
Representative Western blot of the effect of C-phycoerythrin (C-PE) on HgCl_2_-induced endoplasmic reticulum stress through the IRE1α pathway and the attenuation of cell death.

## Data Availability

Publicly available datasets were analyzed in this study. This data can be found here: [https://drive.google.com/file/d/15HqGDpXfEdC6_lv9RdAO3cq8glJocEbF/view?usp=sharing, accessed on 2 October 2021].

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
