# Peer review of "C-phycoerythrin from Phormidium persicinum Prevents Acute Kidney Injury by Attenuating Oxidative and Endoplasmic Reticulum Stress"

_marinedrugs, 2021, doi:10.3390/md19110589_

Round 1

Reviewer 1 Report

Comments:

The manuscript entitled “The nephroprotective activity of C-phycoerythrin purified from 2 Phormidium persicinum is associated with the prevention of 3 oxidative and endoplasmic reticulum stresses in a model of acute kidney injury” reports a C-phycoerythin (C-PE) isolated from Phormidium persicinum and evaluation of nephroprotective action of C-PE by HgCl2 induced AKI due to the oxidative and endoplasmic reticulum stress reducing. C-PE is the marine-derived ingredient widely used in the cosmetic products and food for supplementary due to the radical scavenging and antioxidant activities. It was reported that C-PE has positive effect on the metabolic and toxic injury liver while the function on kidney is unknown. This manuscript states a study with appropriate research design, detailed method, and adequate results as well as the statistical analysis to draw the conclusion. I would recommend this manuscript publish on Marine Drugs once minor concerns responded.

Minor Concerns:

1, Figure 1C, Is the “A562/A260” the typo? Please double check.

Besides, A562 is considered as a characterized absorbance band for determining C-PE in section 2.1. This reviewer is curious about why A562. Is there any study on C-PE characterization? Citation of a C-PE characterization study would help reader get more information about C-PE.

2, Figure 1E, this reviewer is confused with the band corresponding to C-PE (240 KDa mentioned in Line 184) on either native PAGE or SDS PAGE. Since no marker on native page, how to tell readers the desired band? This reviewer would appreciate more details about figure 1E in section 2.1.

3, Line 270, Following the precipitation with ammonia sulfate, which part was dialyzed against distilled water? This description of process in purification should be improved.

4, In addition, this reviewer desires to confirm if distilled water was used as dialysis buffer? Would C-PE be stable in distilled water?

5, This reviewer concern about the form of title. The title should be improved to be concise and in phrase.

6, This reviewer requests the author go though the manuscript carefully with all subscripts of molecular formula. For instance, “HgCl2”, line 257-259, “0.8 g/L NaHCO3, 0.05 g/L K2HPO4, 2.16 g/L NaNO3, 5 mg/L MgSO4, 1 mg/L FeSO4, 1 257 mL of a micronutrient solutions which contain 0.2 mM EDTA, 46.2 mM H3BO3, 9.3 mM 258 MnCl2, 0.95 mM ZnSO4, 2.03 mM Na2MoO4, 0.49 mM Ca(NO3)2, and 0.77 mM CuSO4”.

7, Be careful about the writing in Line 91, “It shows and C-PE subunits corresponding to 19 and 21 kDa”.

Author Response

Dear reviewer, We also appreciate your comments very much after the article revision. We send you the reply of the comments of all the reviewers because we think is better for the complete process. Your revision was number one.

Kind regards.

Reviewer 2 Report

This is an interesting research work on the effect of C-phycoerythrin in preventing oxidative stress, cell damage prevention.

Major English editing is required, for there are many mistakes, making it sometimes hard to understand the text.

The methods seem adequate for the purpose of the paper, although this section could be more clear and should reference the methods.

The results presented are interesting and sound, but the figures need to be improved.

The major problem I see is the discussion, which is too simple, and barely discusses the results, if they are discussed at all.  

Also, the conclusions are just a statement of the main results.

Therefore, although I find the paper very interesting, I believe that the paper needs to be considerably improved prior to publication.

Comments are made below to improve the document.

Abstract:

Line 30 – substitute AKI by acute kidney injury (AKI)

Use the same standard in line 29 and line 31: saline solution = 0,9%SS

Introduction:

As I understood, the pathway that is being investigated is the PERK. I would like this pathway to be described in detail, and not only an enumeration of molecules involved. This would help in the result discussion.

Line 75 – Phormidium should be in italic.

Line 79 - Pseudoanabaena tenuis should be in italic.

Results
2.1. Characterization of C-PE from P. persicinum

Line 86 - State the full scientific name for P. persicinum (Phormidium persicinum)

Line 91 – correct sentence. I’m afraid I cannot see C_PE subunits in figure 1 because it is so small.

Line 98 – separate purification steps (figure 1 A to D) from separation steps (E and F), and improve graphics quality.

Make a new Figure 2 – PAGE

I would advise you to make also a figure 3 for the EEM and explain what we are seeing. What is “A” ? The EEM of C-PE after centrifugation also? Make the caption much more complete.

Line 99 - Phormidium persicinum should be in italic every time.

Line 122 – delete “figure”

Line 132 – graphics in figure 3 are not readable – very small size and low quality. Again, you may separate this into 3 figures: A to E; the photos (F), and the blot of protein expression which is not on the caption.

Line 132 – methods refer to “Kidney damage score” but the results make no reference to these histological scores, and are not discusses at all.

Figures 4 and 5  – again, increase the size of graphics to make them more readable. Separate protein expression blot.

Discussion:

line 169 – correct to “C-PE from Phormidium persicinum”

Discuss the purity obtained for C-PE. Compare e.g., 10.1016/j.jchromb.2015.04.012  

Lines 181-182 – your results are very interesting, but not discussed at all in these two lines. Explain further why C-PE interferes with the oxidative and RE stress metabolism. You never really explained the connection (very briefly in the introduction).  

The discussion is rather poor. explain in detail why your (each) result is important, in which pathway, compare with other studied molecules, ...

Methods:

The official Mexican protocol mentioned if for zoosanitation inspections (which is fine), but no references are cited for the methods used;

 Lines 235 and 241 - Why did you choose 6 mice for the first set of experiments and only 3 for the second?

Lines 244 and 245 - Why did you wait for e days for the second protocol and three days for the first one?

4.2. Cultivation, purification, and characterization of C-PE of Phormidium persicinum.

Line 254 - Phormidium persicinum in italic.

The methods used are not cited and should be. Who developed these protocols= Sfriso et al (2018) are referred, but it is not their protocol, is it?

Line 256 – use the full name of the medium NM (if available).

Line 256 - NaHCO3 and not NaHCO3. The same for the other formulas throughout the text and figures.

Line 281 – why absorbance at 262 nm? https://www.ncbi.nlm.nih.gov/pmc/articles/PMC4614113/

How did you calculate the purity index? https://www.ncbi.nlm.nih.gov/pmc/articles/PMC4614113/

Use citations, if available, for all the methods used.

Line 298 – SDS-Page is the electrophorese technique and thus should be at the end of the sentence. The protein separation methods should be better explained because this is the purpose of these steps: in fact, it is inside “Evaluation of oxidative stress” and it shouldn’t. Make a new subsection for separation steps. This will help to understand figure 1.

Separate the Histological methods.

Conclusions:

The text states solely the results, so these conclusions could be much more interesting.

What are the consequences of such results?

What further work needs to be done?

Author Response

Dear reviewer, We also appreciate your comments very much after the article revision. We send you the reply of the comments of all the reviewers because we think is better for the complete process. Your revision was number two.

Kind regards.

Reviewer 3 Report

The manuscript, entitled “The nephroprotective activity of C-phycoerythrin purified from Phormidium persicinum is associated with the prevention of oxidative and endoplasmic reticulum stresses in a model of acute kidney injury”, demonstrates that purified C-phycoerythrin (C-PE) from P. persicinum protect against HgCl2-caused oxidative stress and cellular damage in the kidney. Mechanistically, mercury activates three pathways of endoplasmic reticulum stress (ERS): PERK/eIF2α/ATF4, ATF6α and IRE1α in acute kidney injury (AKI). As a prodrug, oral gavage administration of C-PE partially reduces ERS by preventing IRE1α signaling pathway and impairs nephrin and podocin disturbance in the model of acute kidney injury induced by HgCl2.

This study reveals the mechanism of the nephroprotective activity of C-PE and results are solid and well organized. However, specific issues suggested for attention include:

Major issues:

  1. In Figure 5 G and I, after administration of 100 mg/Kg/d of C-PE for 3 days, the protein levels of p-p53 and p-p53/p53 ratio also shown significant increase in 0.9% SSI treated mice, does it indicate that high dose of C-PE will induce cell apoptosis in kidney? This issue should be addressed and discussed in the Discussion part.
  2. The rationale of different doses of C-PE administration on mice should be provided in the manuscript.
  3. Have you measured the toxicity of C-PE?
  4. Describe the rationale and results in details for doing various assays in 2.2 and 2.3.

Minor issue:

  1. Reorganize abstract, did not mention C-PE purification and too many details of mouse experiment design.
  2. Some content is missing, leave blank space in the manuscript.
  3. The number of administrations of HgCl2 should be clearly stated.
  4. Indicate the number of specimens for each experiment in Figure legends.
  5. Scale bar was missing in Figure 3F.
  6. Reorganize the WB figure, each sample should correspond well.
  7. In 4.3, delete most of the details of WB assay, briefly describe WB assay, and put in some details of the other assays (line 288-290).

Author Response

Dear reviewer, We also appreciate your comments very much after the article revision. We send you the reply of the comments of all the reviewers because we think is better for the complete process. Your revision was the number three.

Kind regards.

Round 2

Reviewer 2 Report

The changes I suggested were thoroughly made by the authors.

Therefore the paper is much more clear, interesting, and is now fully discussed. Thus, it can be published after minor changes.

For some reason, the format of the paper doesn't fit the layout of “marine drugs”. Thus, it needs to be fully formatted.

Also, the graphs do not have enough resolution.

I also advise the authors to review the text, for the English needs some editing.

Author Response

Dear Reviewer 2, the article was English editing and we modify the format of the article. Currently, the graphs are at the end article one per page to increase the size for review and we converted to ZIP format all the figures in TIF (300 dpi) or JPEG for the graphical abstract and western blot.

Kind regards 

Edgar Cano-Europa 
